# Applying reinforcement learning to optical cavity locking tasks: considerations on actor–critic architectures and real-time hardware implementation

Mateusz Bawaj* and Andrea Svizzeretto

*Department of Physics and Geology, Università di Perugia and INFN, Sezione di Perugia, (PG) Italy*
(Dated: September 19, 2025)

This proceedings contains our considerations made during and after fruitful discussions held at EuCAIFCon 2025. We explore the use of deep reinforcement learning for autonomous locking of Fabry–Perot optical cavities in non-linear regimes, with relevance to gravitational-wave detectors. A custom Gymnasium environment with a time-domain simulator enabled training of agents such as deep deterministic policy gradient, achieving reliable lock acquisition for both low- and high-finesse cavities, including Virgo-like parameters. We also discuss possible improvements with Twin Delayed DDPG, Soft Actor Critic and meta-reinforcement learning, as well as strategies for low-latency execution and off-line policy updates to address hardware limitations. These studies lay the groundwork for future deployment of reinforcement learning-based control in real optical setups.

## INTRODUCTION

Experiments on gravitational wave (GW) are not an exception to the worldwide trend of artificial intelligence (AI) evolution. Current experiments like LIGO, Virgo and KAGRA develop and implement new techniques, mostly focused on data elaboration [1]. The next-generation detector, the Einstein Telescope [2], will benefit from machine learning (ML)-accelerated design optimization [3] and generative AI-aided exploration [4]. We believe that our effort to involve ML in control strategies for experimental setups is the right direction. In these proceedings, we briefly recall our results and describe new ideas born during the EuCAIFCon 2025 (EuCAIFCon).

We attempt a training of an RL agent in a custom Gymnasium environment. The goal is to develop a strategy to autonomously lock a Fabry-Perot (FP) optical cavity operating in a non-linear regime using a deep reinforcement learning (RL) agent, for future applications in GW detectors [5]. The work began with the design and validation of a highly accurate time-domain simulator that models cavity dynamics under varying mirror velocities, including cavity ring-down effects at high finesse. This simulator is embedded into a custom Gymnasium environment, enabling the RL agent to interact with realistic cavity conditions.

The core control task, locking the cavity, is achieved by training a deep deterministic policy gradient (DDPG) agent [6], which continuously adjusts mirror positions to reach and maintain resonance. We engineered a reward function based on cavity power and the Pound-Drever-Hall (PDH) error signal to guide the learning process [5]. A carefully tuned reward function prevents the agent from learning oscillatory or suboptimal behaviours.

The agent is tested on both low- and high-finesse cavities, including those matching the parameters of the Virgo interferometer, demonstrating reliable lock acquisition even under strong nonlinearity. This work is currently being expanded to address the Sim2Real challenge by analysing delays and random effects and modifying the environment accordingly, with promising results in noisy scenarios. Partial observability and domain randomization are reported in a separate proceedings contribution. This work lays the foundation for future deployment of RL-driven control in real optical setups.

## IMPROVEMENTS TO THE ACTOR-CRITIC ARCHITECTURE

Despite the DDPG algorithm giving us the best results during training, there are more advanced successors worth our attention: Twin Delayed DDPG (TD3) [7], known for its more accurate policy updates, and Soft Actor Critic (SAC), which enables better exploration of the action space. In many tasks, both outperform DDPG in providing more stable policies thanks to the use of two critic networks [7–9]. The evolution of DDPG was mainly discussed during the meeting; however, other, mostly application-focused studies [10, 11], reveal that there is no clear winner between them and indicate Proximal Policy Optimization (PPO) [12] as the best-performing deep reinforcement learning (DRL) algorithm in certain tasks. Discussions during EuCAIFCon and the literature review motivate us to carry out thorough testing of all candidates in our environment.

Further, we discussed a possible evolution of our system toward meta reinforcement learning (Meta-RL) [13]. This approach would shift the DRL algorithm design problem to another level. Instead of manually tuning the algorithm, we would incorporate the tuning process into training in order to achieve better generalization, adaptability and efficiency across a wide spectrum of tasks.

arXiv:2509.14884v1 [physics.ins-det] 18 Sep 2025

Summarizing possible strategies, we think that currently Meta-RL training is computationally too heavy for us and that the risk of producing algorithms unstable in implementation is too high. We therefore concluded that our current system will benefit from a temporal unit (recurrent neural network (RNN) or gated recurrent unit (GRU)). Adding such a layer helps extract information from order-sensitive sequential data. We will focus on this development first.

## LOW-LATENCY EXECUTION

In feedback time delay systems (TDSs), the key aspect is the reaction time of the controller. Relevant contributions in digital controllers come from: sensor, analogue-to-digital converter (ADC), computing, digital-to-analogue converter (DAC) and actuator propagation time. Our system differs from a classic digital signal processor (DSP) only in the computing algorithm. Therefore, we will focus only on the RL-agent inference computing time, known bottlenecks and possible techniques to improve it. As a reference for the data processing delay, we take two systems used to control the Virgo experiment [14]: the DSP for super-attenuator control, which guarantees a $3.125\,\mu s$ delay, and the real-time DAQ, which guarantees $100\,\mu s$. Neither of the two systems is ready to accelerate ML-agent inference. Therefore, our current setup uses a Jetson Nano, which computes DDPG actor inference in $1.20 \pm 0.87\,ms$. The same chip is responsible for communication with the multichannel ADC (ADS1256) and the DAC (DAC8532). The bottleneck in this case is the actuator, which can be updated at a maximum rate of $200\,Hz$.

Discussions inspired by the work described in [15] brought us subsequent ideas to address the above issues. We can apply two techniques to mitigate the low-frequency DAC update limitation. The simplest solution to describe is to change the hardware. Our constraint on fast sampling and fast inference is naturally associated with the implementation of the agent into field programmable gate array-based electronics. The high-level synthesis language for machine learning HLS4ML [16] will be a natural choice for applying this solution. However, an alternative strategy has attracted our interest. We can lower the RL-agent action rate to a level sustainable by the hardware while keeping the ADC acquisition rate high. According to experience from other applications, this could help the agent to forecast more accurate actions. In fact, we have noticed that an excessive sampling rate of our simulated environment does not bring benefits to the training.

The scarce resources of the Jetson Nano also make in-system policy optimization challenging. We find the possibility of off-line policy updates encouraging. In this case, the on-line agent is pre-trained and left running in the system. Periodically, data acquired during agent operation is transferred to an external computer, which performs policy optimization. The on-line system must then stop, reload the updated policy and resume operation. We plan to implement off-line policy updates as soon as we transfer our system to the real setup.

## CONCLUSION

During EuCAIFCon we reviewed our progress on applying reinforcement learning to the Fabry–Perot cavity locking problem and identified promising directions for further work. DDPG agents trained in a custom Gymnasium environment demonstrated reliable lock acquisition and discussions highlighted possible improvements with TD3, SAC and meta-reinforcement learning. We also addressed the challenges of low-latency execution on embedded hardware and proposed mitigation strategies, including field programmable gate array (FPGA) implementation and off-line policy updates. These considerations strengthen the foundation for transferring our approach from simulation to real optical setups, with the long-term goal of integrating RL-driven control into gravitational-wave experiments.

## ACKNOWLEDGEMENTS

We would like to express our sincere gratitude to Verena Kain and Michael Schenk from CERN for their fruitful discussions and valuable advice on alternatives to DDPG in the context of our real-time control application. Their insights have been highly beneficial in shaping the direction of this work. We would also like to thank Andrea Santamaria Garcia from the University of Liverpool for insightful discussions and valuable advice regarding online policy updates in our system.

*Funding information*  Research is supported by Italian Ministry of University and Research (MUR) through the program "Dipartimenti di Eccellenza 2023-2027" (Grant SUPER-C), Italy.

* mateusz.bawaj@unipg.it

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
