# Peer review of "Applying reinforcement learning to optical cavity locking tasks: considerations on actor-critic architectures and real-time hardware implementation"

_SciPost Physics Proceedings_

## Round 1 · Referee Report · Anonymous (Referee 1) · 2025-12-11

Report

This is a nice contribution to the EUCaifCon proceedings. The tone of the article is somewhat conversational and emphasizes the results of discussions at the meeting. I support this choice of the authors and would largely keep it. However perhaps it can be reduced somewhat to also fit in at least one concrete result from their original talk at the conference.

Some concrete comments:

Intro:
Reference for Gymnasium enviromnet?

"The agent is tested on both low- and high-finesse cavities, including those matching the parameters of the Virgo interferometer, demonstrating reliable lock acquisition even under strong nonlinearity.": This claim might benefit from a figure or table to illustrate the achieved performance

Improvements:
"computationally too heavy for us" -> can there be put a number next to this? What is the binding factor? eg: "would require a 100x increase in training time" could be interesting for the reader, if such a statement can be made

Recommendation

Ask for minor revision

---

## Editorial Decision

awaiting_resubmission